# Adversarially Robust Learning with Uncertain Perturbation Sets

**Tosca Lechner**
University of Waterloo, Waterloo, Ontario, Canada, `tlechner@uwaterloo.ca`

**Vinayak Pathak**
Layer 6 AI, Toronto, Ontario, Canada, `vinayak@layer6.ai`

**Ruth Urner**
York University, Toronto, Ontario, Canada, `uruth@eecs.yorku.ca`

## Abstract

In many real-world settings exact perturbation sets to be used by an adversary are not plausibly available to a learner. While prior literature has studied both scenarios with completely known and completely unknown perturbation sets, we propose an in-between setting of learning with respect to a class of perturbation sets. We show that in this setting we can improve on previous results with completely unknown perturbation sets, while still addressing the concerns of not having perfect knowledge of these sets in real life. In particular, we give the first positive results for the learnability of infinite Littlestone classes when having access to a perfect-attack oracle. We also consider a setting of learning with abstention, where predictions are considered robustness violations, only when the wrong label prediction is made within the perturbation set. We show there are classes for which perturbation-set unaware learning without query access is possible, but abstention is required.

## 1 Introduction

Adversarial perturbations, imperceivably small manipulations to input instances that cause unexpected failures of learned predictors, pose significant safety concerns for their employment. The phenomenon, first exposed a decade ago in image classification tasks [Szegedy et al., 2014], has since received substantial attention, both in the practical and theoretical machine learning research communities. Studies in both lines of research often skip over the question of how to suitably model "imperceivable perturbations" and rather investigate how to defend against various (fixed) types of attacks. In theoretical studies, this is typically modeled by defining an *adversarial loss* that, in addition to misclassification, penalizes the existence of a perturbed instance on which a classifier assigns a different label. This type of loss definition then crucially depends on a given *perturbation type*, that is some function $u : \mathcal{X} \to 2^{\mathcal{X}}$ that assigns every domain instance a set of points that would be viewed as an admissible perturbation (often balls of some radius with respect to some norm are considered).

However, in most realistic settings, exact perturbation sets to be used by an adversary are not plausibly available to a learner. Moreover, it has been shown that encouraging robustness with respect to a perturbation type that is not suitable for the task easily degrades classification accuracy [Tsipras et al., 2019, Zhang et al., 2019, Yang et al., 2020]. In response, some work has then explored the question of learnability when the perturbation type is not known at all in advance [Montasser et al., 2021], but information about possible perturbations can be accessed through queries. It is not difficult to be convinced though that without any restrictions or knowledge of the perturbations to be used, robust

37th Conference on Neural Information Processing Systems (NeurIPS 2023).

learning from samples only is impossible. Even for a small collection of possible perturbation types, as soon as these induce inconsistencies between optimal predictors a sample based learner cannot be simultaneously successful with respect to all options.

In this work we explore a middle ground between fixed and unknown perturbation types, which we term *learning with uncertain perturbation sets*. We will assume that the true perturbation type is a member of a fixed class of perturbation types. One can think of this as knowing that we should encourage robustness with respect to some type of norm induced ball, but may not be certain about which type of norm or which radius is most suitable for the task. We then study which structures (on the interplay between perturbation type class and the class of predictors to be learned) allow for PAC-type learnability guarantees.

Given a class $\mathcal{U}$ of perturbation types and hypothesis class $\mathcal{H}$, we define an $\mathcal{H}$-induced partial order on $\mathcal{U}$. We show that when $\mathcal{U}$ is actually totally ordered in this sense, statistical (PAC) learning is possible in the realizable case as soon as $\mathcal{H}$ has finite VC-dimension. $\mathcal{U}$ being totally ordered applies to a wide variety of settings, for example the case where the various perturbation types are balls in some metric space while the radius varies over $\mathcal{U}$ (and actually also does not need to be constant over domain $\mathcal{X}$ for each $u \in \mathcal{U}$). However, we also show that, without the realizability assumption learning from samples alone is not feasible, even when $\mathcal{U}$ is totally ordered with respect to $\mathcal{H}$.

We thus explore two natural remedies: (1) We allow the learner to interact with a *perfect attack oracle* [Montasser et al., 2021]. Given an instance $x$ and classifier $f$, such an oracle certifies robustness or provides an adversarial perturbation of $x$ for $f$. Previous work on learning with unknown perturbation sets has employed such an oracle and shown learnability for classes with finite Littlestone dimension [Montasser et al., 2021]. In this work, we show that provided certain structures ($\mathcal{U}$ being totally ordered or a finite union of totally ordered perturbation type classes) learning with a perfect attack oracle becomes feasible for all classes of finite VC-dimension (which is a much wider space of natural hypothesis classes than those of finite Littlestone dimension). (2) We allow the learner to output a classifier that sometimes *abstains from prediction*. For this, we define a modification of the adversarial loss, where a hypothesis does not suffer loss when it abstains on an instance that was actually manipulated. We show that this again yields learnability for VC-classes when the perturbation type class is a finite union of totally ordered classes. We then consider the case in which $\mathcal{H}$ has finite disagreement-coefficient [Hanneke, 2007] and show that such $\mathcal{H}$ can be learned with respect to every class of perturbation types. And finally, we define a notion of $(\epsilon, \mathcal{H})$-cover for a class of perturbation types $\mathcal{U}$, introduce learners for this kind of cover and show that a class $\mathcal{H}$ is robustly learnable with respect to $\mathcal{U}$ if there is a finite-disagreement-cover learner for $(\mathcal{H}, \mathcal{U})$.

Our results on learning with abstentions show that different strategies for guarantees for different kinds of robustness (for example different types of perturbation sets) can be combined into an abstention learner that only predicts in the agreement region. The guarantees for each learning strategy then yield a guarantee for the combined classifier that depends on the number of strategies used. This highlights a different aspect to the power of abstention in adversarially robust learning from what has been established in prior work [Balcan et al., 2020, 2023]. We show that abstentions can also be used to address uncertainty in perturbation types.

## 1.1 Related work

Providing PAC type learning guarantees under adversarial robustness requirements has received an enormous amount of research attention in the past decade [Feige et al., 2015, Cullina et al., 2018, Wang et al., 2018, Awasthi et al., 2019, Montasser et al., 2019, Attias et al., 2019, Montasser et al., 2020, Ashtiani et al., 2020, Montasser et al., 2021, Gourdeau et al., 2021, 2022, Montasser et al., 2022, Attias and Hanneke, 2022, Attias et al., 2022, Bhattacharjee et al., 2021, Awasthi et al., 2022b,a, 2023, Mao et al., 2023]. We will focus here on aspects within this literature that are most relevant to our settings.

Most prior works study the sample complexity of adversarial robustness with respect to a fixed type of adversarial perturbation (often a metric ball with a certain radius). However, recent research has developed frameworks of analysis that go beyond learning with respect to a fixed type of perturbations. It has been argued that the "correct" type of admissible adversarial perturbations (which originally were "imperceivable" modifications to input instances that lead to misclassification by a learned predictor) should depend on the underlying data-generating process [Bhattacharjee and Chaudhuri,

2021, Chowdhury and Urner, 2022]. These studies define a notion of "adaptive robustness" with respect to which a predictor should be evaluated. The drawback of these notions is that the correct perturbation type is not available to the learner and thus a predictor cannot be straightforwardly evaluated with respect to this loss. A different way of relaxing the assumption of a fixed, known perturbation type is provided in the framework of *tolerance* [Ashtiani et al., 2023, Bhattacharjee et al., 2023]. Here, a learner is evaluated with respect to some fixed perturbation type, while being compared to the optimal achievable loss with respect to a larger perturbation type, relaxing the requirement for the learner and reflecting that the exact relevant perturbation type is typically not known (or even uniquely existing).

Most relevant to our work is a recent study on PAC learning with respect to *unknown perturbation sets* [Montasser et al., 2021]. This work provides learning guarantees for the case that the robustness requirement will be with respect to an entirely unknown perturbation type without any prior knowledge about the nature of these perturbations. Those learning bounds assume access to a perfect attack oracle (as we do in some parts of this work), and require the class to have finite Littlestone dimension. In this work, we show that adding some structure to the class of perturbation types (as well as the promise that the true perturbation type will be a member of this class), allows for sample and query complexity bounds that are independent of the hypothesis class's Littlestone dimension.

Finally, there is a long line of work studying the benefits of abstentions for classification tasks [Bartlett and Wegkamp, 2008, El-Yaniv and Wiener, 2010, Wiener and El-Yaniv, 2015, Yuan and Wegkamp, 2010, Zhang and Chaudhuri, 2016]. Some recent studies have also shown that the ability to abstain from prediction for manipulated instances can be beneficial for adversarial robustness requirements [Balcan et al., 2020, 2023]. However, the former studies a different setting for the generation of adversarial perturbations, and the latter considers fixed perturbation types.

## 1.2 Overview of results

In Section 2, we define our setup, introduce the notions of learnability and loss we investigate and provide an overview of the different kinds of prior knowledge and oracle access we employ. In particular, we there define our partial order on classes of perturbation types.

In Section 3 we analyze perturbation type classes for which this order is a total order. Theorem 1 states that every hypothesis class with finite VC dimension can be robustly learned in the realizable case with respect to any totally ordered perturbation type class. We then prove that this result cannot be extended to the agnostic case (Observation 1). This motivates investigating the benefits of a perfect attack oracle. Theorem 2 shows that the impossibility can be overcome with access to a perfect attack oracle and provides a finite sample and oracle-query bound for the agnostic case.

In Section 4, we investigate classes of perturbation types $\mathcal{U}$ that are not necessarily totally ordered. We first extend our previous results on learning with access to a perfect-attack-oracle to the case where $\mathcal{U}$ is a union of finitely many totally ordered perturbation classes (Theorem 3). We then establish that there are classes of perturbation types and hypothesis classes which cannot be learned without abstention, even with access to a perfect attack oracle (Observation 3), but which can be learned with abstention without the need for access to a perfect-attack-oracle (Observation 4). This motivates our investigations into learning with respect to our adversarial abstention loss (Definition 5).

We show that in the realizable case, every hypothesis class with finite VC-dimension can be learned with respect to a finite union of totally ordered perturbation types in Theorem 4. We then consider the case in which $\mathcal{H}$ has finite disagreement-coefficient and show that such $\mathcal{H}$ can be learned with respect to every class of perturbation types (Theorem 5). Lastly, we define a notion of $(\epsilon, \mathcal{H})$-cover for a class of perturbation types $\mathcal{U}$, which states that for every $u \in U$ there is a close-to-optimal hypothesis for $u$ in the cover (Definition 7) and introduce the notion of learners for this kind of cover (Definition 8). We then generalize our previous results by showing that a class $\mathcal{H}$ is robustly learnable with respect to $\mathcal{U}$ if there is a finite-disagreement-cover learner for $(\mathcal{H}, \mathcal{U})$ (Theorem 6). While we provide some proof sketches in the main body of the paper, the detailed proofs for all results are in the supplement.

## 2 Setup

For sets $A$ and $B$, we let $2^A$ denote the powerset of $A$ and $A^B$ the set of all functions from $B$ to $A$. Let $\mathcal{X}$ be a domain and $\mathcal{Y} = \{0, 1\}$ be a label space. We model the data generation as a distribution

$P$ over $\mathcal{X} \times \mathcal{Y}$, and let $P_\mathcal{X}$ denote its marginal over $\mathcal{X}$. A *hypothesis* or *predictor* is a function $h : \mathcal{X} \to \{0, 1, \star\}$. Thus, on input $x \in \mathcal{X}$, a hypothesis $h$ either predicts label 0 or 1 or abstains by outputting $\star$. We call a hypothesis *non-abstaining* if $h(x) \neq \star$ for all $x \in \mathcal{X}$. A *hypothesis class* $\mathcal{H}$ is a set of non-abstaining predictors. We will let $\mathcal{F} = \{0, 1, \star\}^\mathcal{X}$ denote the set of all predictors over $\mathcal{X}$.

The performance of a predictor $h$ is measured by a loss function $\ell : \mathcal{F} \times \mathcal{X} \times \mathcal{Y} \to \mathbb{R}$. For loss $\ell$, we use the notation $\mathcal{L}_P(h) = \mathbb{E}_{(x,y) \sim P}[\ell(h, x, y)]$, to denote the *expected loss* of predictor $h$ with respect to distribution $P$ and $\mathcal{L}_S(h) = \frac{1}{|S|} \sum_{i=1}^{|S|} \ell(h, x_i, y_i)$ the *empirical loss* on dataset $S = ((x_1, y_1), \ldots, (x_m, y_m))$. Further, for a hypothesis class $\mathcal{H}$ we will let $\mathrm{opt}_P(\mathcal{H}) = \inf_{h \in \mathcal{H}} \mathcal{L}_P(h)$ denote the *approximation error* of class $\mathcal{H}$ with respect to $P$ and loss $\ell$.

The standard loss for classification tasks is the binary loss $\ell^{0/1}(h, x, y) = \mathbb{1}[h(x) \neq y]$, where $\mathbb{1}[\cdot]$ denotes the indicator function. The standard loss for adversarial robustness additionally penalizes $h$ on instance $(x, y)$ for the existence of a perturbation on $x$ that $h$ does not label with $y$. This loss is defined for a specified *perturbation type*, which is a function $u : \mathcal{X} \to 2^\mathcal{X}$ that assigns every instance $x \in \mathcal{X}$ a *set* $u(x) \subseteq \mathcal{X}$ with $x \in u(x)$ of *admissible perturbations*. The *adversarial loss* with respect to $u$ is then defined as

$$\ell^u(h, x, y) = \mathbb{1}[\exists z \in u(x) : h(z) \neq y].$$

We will use the notation $\ell^{u,\perp}(h, x)$ to denote the *adversarial component loss*, which indicates whether a domain point $x$ lies close to the decision boundary with respect to the perturbation type $u$:

$$\ell^{u,\perp}(h, x) = \mathbb{1}[\exists z \in u(x) : h(z) \neq h(x)].$$

For hypotheses that have the option to abstain from prediction, we propose a variation of the adversarial loss that allows for a predictor to abstain from prediction on perturbed instances (but not on unperturbed instances) without suffering loss. The following definition captures this modification:

**Definition 1** (Adversarial abstention loss). *For a perturbation type $u : \mathcal{X} \to 2^\mathcal{X}$, the* adversarial abstention loss *is defined by*

$$\ell^{u,\mathrm{abst}}(h, x, y) = \begin{cases} 1 & \textit{if } h(x) \neq y \\ 1 & \textit{if } \exists z \in u(x) \textit{ such that } h(z) \neq y \textit{ and } h(z) \neq \star \\ 0 & \textit{otherwise} \end{cases}$$

The main difference with adversarial loss is that if $h(z) = \star$ for an adversarial point $z \in u(x)$ with $z \neq x$, then there is no penalty and $\ell^{u,\mathrm{abst}}(h, x, y) = 0$, thus allowing the predictor to abstain on points that were perturbed. This definition models the scenario where if the predictor can correctly detect that an adversarial attack has happened and abstains, then it is not penalized. However, abstaining on non-perturbed points is still penalized. Note that in case a predictor $h$ never abstains (that is $h(x) \neq \star$ for all $x \in \mathcal{X}$) the two definitions of adversarial loss coincide. We let $\mathcal{L}_P^u(h)$, $\mathcal{L}_P^{u,\mathrm{abst}}(h), \mathcal{L}_S^u(h), \mathcal{L}_S^{u,\mathrm{abst}}(h)$ denote the corresponding expected and empirical losses, and $\mathrm{opt}_P^u(\mathcal{H})$, $\mathrm{opt}_P^{u,\mathrm{abst}}(\mathcal{H})$ the corresponding approximation errors of a class $\mathcal{H}$ and distribution $P$.

**Uncertain perturbation type** In the literature, adversarial robustness is mostly treated with respect to a fixed (known) perturbation type $u$, while learning with respect to an entirely unknown perturbation type has also been investigated. In this work, we introduce a setting that naturally interpolates between these two extremes. We assume that the true perturbation type $u^*$ is unknown to the learner, but promised to lie within a fixed *class $\mathcal{U}$ of perturbation types*. That is, the learner has prior knowledge of a class $\mathcal{U}$ with $u^* \in \mathcal{U}$, where $u^*$ is the perturbation type that the learner will be evaluated with (see Definition 3 below). We can think of $u^*$ as the perturbation type that the true adversarial environment will employ to manipulate input instances and induce misclassifications. We let $\mathcal{U}^{\mathrm{all}}$ denote the class that contains all possible perturbation types.

It is easy to see that without any restrictions on the class $\mathcal{U}$ and any capability of the learner to (interactively) gather information about $u^*$ and no option to abstain, a learner cannot succeed (see also Observations 1 and 2 below). To gain control over potentially infinite classes of perturbation types, we will now define a partial order on a class $\mathcal{U}$ that is induced by a hypothesis class $\mathcal{H}$. This structure will prove to be conducive for learnability.

**Definition 2.** *Let $\mathcal{H} \subseteq \{0,1\}^{\mathcal{X}}$ be a hypothesis class. For perturbation types $u_1, u_2$ we say that $u_1$ is smaller than $u_2$ with respect to class $\mathcal{H}$, and write $u_1 \preceq_{\mathcal{H}} u_2$, if and only if for all $h \in \mathcal{H}$ and all $x \in \mathcal{X}$ we have*

$$\ell^{u_1, \perp}(h, x) \leq \ell^{u_2, \perp}(h, x).$$

*We say a set of perturbation types $\mathcal{U}$ is* totally ordered, *with respect to $\preceq_{\mathcal{H}}$, if for every $u_1, u_2 \in \mathcal{U}$, we have either $u_1 \preceq_{\mathcal{H}} u_2$ or $u_2 \preceq_{\mathcal{H}} u_1$.*

We have $u_1 \preceq_H u_2$ if, for every function $h \in \mathcal{H}$ the margin area (the points which receive positive adversarial component loss) with respect to $u_1$ is included in the margin area with respect to $u_2$. This is, for example, the case if all perturbation sets with respect to $u_1$ are included in those induced by $u_2$.

**Resources to learners and learnability** A *learner* is a function $A : (\mathcal{X} \times \mathcal{Y})^* \to \mathcal{F}$ that takes in a finite sequence $S = ((x_1, y_1), \dots (x_m, y_m))$ of labeled examples and outputs a predictor $A(S) : \mathcal{X} \to \{0, 1, \star\}$. We will study the following PAC-type [Valiant, 1984] notion of learnability of hypothesis classes $\mathcal{H}$ with perturbation type classes $\mathcal{U}$.

**Definition 3** (Learnability without abstentions). *Let $\mathcal{H} \subseteq \{0,1\}^{\mathcal{X}}$ be a hypothesis class and $\mathcal{U}$ a class of perturbation types. We say that $\mathcal{H}$ is robustly learnable with respect to perturbation class $\mathcal{U}$ if there exists a learner $A$ and a function $m : (0,1)^2 \to \mathbb{N}$ such that for every $\epsilon, \delta > 0$, every $m \geq m(\epsilon, \delta)$, every $u^* \in \mathcal{U}$ and every data-generating distribution $P$ we have*

$$\mathbb{P}_{S \sim P^m}[\mathcal{L}_P^{u^*}(A(S)) \leq \mathrm{opt}_P^{u^*}(\mathcal{H}) + \epsilon] \geq 1 - \delta.$$

*We speak of* learnability in the realizable case *if the above requirement is relaxed to only hold for all distributions $P$ with $\mathrm{opt}_P^{u^*}(\mathcal{H}) = 0$. Without this assumption, we also speak of* learnability in the agnostic case.

If the above definition is fulfilled for a class $\mathcal{H}$, then the function $m : (0,1)^2 \to \mathbb{N}$ in the above definition is an upper bound on the *sample complexity* of the learning task. Note that, while we don't require the learner in the above definition to output a non-abstaining hypothesis, predicting $\star$ will always cause loss with respect to $\ell^u$. We can thus assume all considered predictors are non-abstaining classifiers for this setting.

Note that the above definition generalizes and unifies previous notions of adversarially robust learnability. We obtain learnability with respect a *known perturbation type* (the setting that is mostly considered in prior works) when $\mathcal{U} = \{u^*\}$ is a singleton class. Learning with *unknown perturbations* [Montasser et al., 2021] on the other hand, is the setting where $\mathcal{U}$ is the set of all possible perturbation types. In this work, we are interested in studying more fine-grained conditions on the structure of (the combination of) $\mathcal{H}$ and $\mathcal{U}$ that allow for learnability. We will call the case where $\mathcal{U}$ is neither a singleton set nor the set of all possible perturbation types *learning with uncertain perturbation sets*.

We will show that, even when the class of perturbation types $\mathcal{U}$ in consideration is totally ordered with respect to hypothesis class $\mathcal{H}$, agnostically learning $\mathcal{H}$ with $\mathcal{U}$ is impossible even in very simple cases (see Observation 1). We will thus, as has been done in earlier studies [Ashtiani et al., 2020, Montasser et al., 2021] in addition allow the learner access to a *perfect attack oracle*, which we model as follows:

**Definition 4** (Perfect Attack Oracle). *A perfect attack oracle for perturbation type $u$ is a function $\mathcal{O}_u : \mathcal{F} \times \mathcal{X} \to \mathcal{X} \cup \{\text{robust}\}$, that takes as input a non-abstaining predictor $f$ and a domain point $x$ and either certifies that $f$ is robust on $x$ with respect to $u$ by outputting* robust *or returns an admissible perturbation of $x$ for $f$. That is*

$$\mathcal{O}_u(f, x) = \begin{cases} \text{robust } \textit{if } f(z) = f(x) \textit{ for all } z \in u(x) \\ z \in u(x) \textit{ with } f(x) \neq f(z) \textit{ otherwise} \end{cases}$$

We say that a class $\mathcal{H}$ and perturbation class $\mathcal{U}$ are *learnable with access to a perfect attack oracle* if there exists a function $n : (0,1)^2 \to \mathbb{N}$ such that the conditions in Definition 3 are satisfied for a learner that additionally makes at most $n(\epsilon, \delta)$ queries to the perfect attack oracle. The function $n(\cdot, \cdot)$ then is an upper bound on the attack *oracle complexity* of the learning problem.

In Section 4 we will consider more general pairs of hypothesis and perturbation classes $\mathcal{H}$ and $\mathcal{U}$. We will show that if $\mathcal{U}$ is not necessarily totally ordered with respect to $\mathcal{H}$ (or a finite union of such totally

ordered classes) there are tasks on which no learner (even with access to a perfect attack oracle) can succeed (see Observation 3 below). For such cases we will explore learnability with respect to the adversarial abstention loss:

**Definition 5** (Learnability with abstentions). *Let $\mathcal{H} \subseteq \{0,1\}^{\mathcal{X}}$ be a hypothesis class and $\mathcal{U}$ a class of perturbation types. We say that $\mathcal{H}$ is robustly learnable with abstentions* with respect to perturbation class $\mathcal{U}$ *if there exists a learner $A$ and a function $m : (0,1)^2 \to \mathbb{N}$ such that for every $\epsilon, \delta > 0$, every $m \geq m(\epsilon, \delta)$, every $u^* \in \mathcal{U}$ and every data-generating distribution $P$ we have*

$$\mathbb{P}_{S \sim P^m}[\mathcal{L}_P^{u^*, \mathrm{abst}}(A(S)) \leq \mathrm{opt}_P^{u^*}(\mathcal{H}) + \epsilon] \geq 1 - \delta.$$

Note that since $\mathcal{H}$ contains only non-abstaining predictors, we have $\mathrm{opt}_P^{u^*}(\mathcal{H}) = \mathrm{opt}_P^{u^*, \mathrm{abst}}(\mathcal{H})$.

**Discussion on additional parameters and assumptions**    Throughout this paper, we will also work with various standard complexity measures, such as bounded VC-dimension or bounded Littlestone dimension of the hypothesis class $\mathcal{H}$, bounded VC-dimension of the loss class or the adversarial component loss class. We refer the reader to the appendix for a reminder of the definitions of these notions.

We will further assume that learners have the capability to identify empirical risk minimizing hypotheses from a class. It is standard (though often implicit) to assume this as oracle access for the learner as well. We consider the standard ERM oracle, a robust ERM (RERM) oracle (for a fixed perturbation type $u$) and a maximally robust ERM (MRERM) oracle (for a class $\mathcal{U}$ of perturbation types, and realizable samples only). We define the following oracles:

$$\mathrm{ERM}_{\mathcal{H}} : S \mapsto h_S \in \mathrm{argmin}_{h \in \mathcal{H}} \mathcal{L}_S^{0/1}(h)$$
$$\mathrm{RERM}_{\mathcal{H}}^{\mathrm{u}} : S \mapsto h_S^u \in \mathrm{argmin}_{h \in \mathcal{H}} \mathcal{L}_S^u(h)$$
$$\mathrm{MRERM}_{\mathcal{H}}^{\mathcal{U}} : S \mapsto \begin{cases} (h_S^{u_S^*}, u_S^*) \text{ with } h_S^{u_S^*} \in \mathrm{argmin}_{h \in \mathcal{H}} \mathcal{L}_S^{u_S^*}(h), \\ \qquad \text{and } u_S^* \text{ is } \preceq_{\mathcal{H}} \text{ maximal in the set } \{u \in \mathcal{U} \mid \min_{h \in \mathcal{H}} \mathcal{L}_S^u(h) = 0\} \\ \text{error if } \mathcal{L}_S^u(h) > 0 \text{ for all } u \in \mathcal{U} \text{ and } h \in \mathcal{H} \end{cases}$$

That is, we will assume that $\mathrm{MRERM}_{\mathcal{H}}^{\mathcal{U}}$ will return an error if the input sample $S$ is not $\ell^u$-realizable for any $u \in \mathcal{U}$. If, on the other hand, there does exist a $u \in \mathcal{U}$ for which $S$ is $\ell^u$-realizable, then $\mathrm{MRERM}_{\mathcal{H}}^{\mathcal{U}}$ will return a maximal (with respect to $\preceq_{\mathcal{H}}$) such perturbation type $u_S^*$ and corresponding hypothesis $h_S^{u_S^*}$ with $\mathcal{L}_S^{u_S^*}(h_S^{u^*}) = 0$. In particular, we assume that for every non-empty sample $S$ there exists such a maximal element $u_S^* \in \mathcal{U}$ and corresponding ERM hypothesis from $\mathcal{H}$. While these do not always exist in $\mathcal{U}$ and $\mathcal{H}$ a priori, it has recently been shown that it is possible to embed $\mathcal{U}$ and $\mathcal{H}$ into slightly larger classes so that the $\mathrm{MRERM}_{\mathcal{H}}^{\mathcal{U}}$ oracle is always well defined [Lechner et al., 2023]. See Appendix Section A for more details on this.

Since we focus on studying sample complexity (independently of computational considerations) and consider learners as functions throughout, assuming access to the above oracles is not restricting the validity of our bounds. It is still interesting to understand when is their existence reasonable to assume. Both ERM and RERM oracles have been widely used in the literature and they are often computationally hard to implement. Assuming their existence, however, an MRERM oracle can be easily implemented for the case where $\mathcal{U}$ is finite and totally ordered by running a binary search over $\mathcal{U}$ and calling RERM for the comparison step. If we assume $\mathcal{U}$ is parametrized by a $k$-bit number, then this requires $O(k)$ queries to RERM.

## 3    Learning with a totally ordered perturbation class

We will start with investigating the case where the perturbation class $\mathcal{U}$ is totally ordered with respect to hypothesis class $\mathcal{H}$ (see Definition 2). Note that if two perturbation types $u_1$ and $u_2$ we satisfy $u_1(x) \subseteq u_2(x)$ for all $x \in \mathcal{X}$, then $u_1$ is smaller than $u_2$, $u_1 \preceq_{\mathcal{H}} u_2$, for every hypothesis class $\mathcal{H} \subset \{0,1\}^{\mathcal{X}}$. This is, for example, the case when $u_1$ and $u_2$ assign balls (with respect to some metric) centered at $x$ to every domain point $x$ and $u_2$ always assigns a ball of radius at least as large as the ball assigned by $u_1$ (while the balls assigned by $u_1$ and $u_2$ are not necessarily required to have the same radii uniformly over the space $\mathcal{X}$).

In this section, we will focus on learning with respect to the standard adversarial loss and thus only consider non-abstaining predictors. We first show that for $\mathcal{H}$ and totally ordered $\mathcal{U}$, whenever the class $\mathcal{H}$ has bounded VC-dimension we get learnability in the realizable case. This result is based on adapting a compression argument for adversarially robust learning [Montasser et al., 2019] to the case of uncertain perturbation sets. A very similar adaptation has recently been made for the related problem of strategic classification [Lechner et al., 2023]. The proof of this result can be found in the appendix.

**Theorem 1.** *Let $\mathcal{U}$ be a perturbation class that is totally ordered with respect to a hypothesis class $\mathcal{H}$ with $\mathrm{VC}(\mathcal{H}) = d < \infty$. Then $\mathcal{H}$ is learnable in the realizable case with respect to $\mathcal{U}$ with sample complexity $O(\frac{d \cdot 2^d \cdot \log(1/\delta)}{\epsilon})$.*

Recall that realizability here means $\mathrm{opt}_P^{u^*}(\mathcal{H}) = 0$ for the true perturbation type $u^*$ (with respect to which the learner is evaluated). We now show that without this assumption learning guarantees are impossible without equipping the learner with additional resources (or weakening the success criteria).

**Observation 1.** *There exists a class $\mathcal{H}$ with $\mathrm{VC}(\mathcal{H}) = 1$ and a perturbation class $\mathcal{U}$ that is totally ordered with respect to $\mathcal{H}$ that is not learnable (in the sense of Definition 3).*

We will now show that allowing a learner access to a perfect attack oracle yields learnability in the agnostic case.

**Theorem 2.** *Let $\mathcal{U}$ be a perturbation class that is totally ordered with respect to a hypothesis class $\mathcal{H}$ with $\mathrm{VC}(\mathcal{H}) = d < \infty$ and assume access to a perfect attack oracle. Then $\mathcal{H}$ is learnable in the agnostic case with respect to $\mathcal{U}$ with sample complexity $m(\epsilon, \delta) = O(\frac{d \cdot 2^d \cdot \log(1/\delta)}{\epsilon^2})$ and oracle complexity $O(m(\epsilon, \delta)^2)$.*

*Proof.* We will employ a well-known reduction from agnostic learning to realizable compression-based learning [Montasser et al., 2019]. Since our proof for Theorem 1 for the realizable case of learning with uncertain, totally ordered perturbation sets employs a compression-based learner, for the agnostic case it suffices to show that, given any sample $S = ((x_1, y_1), \ldots, (x_m, y_m))$, we can identify a *largest* subsample $S'$ of $S$ that is realizable with respect to the underlying perturbation type $u^*$. We will now outline how the perfect attack oracle can be employed to achieve this.

Employing the $\mathrm{MRERM}_{\mathcal{H}}^{\mathcal{U}}$-oracle, the learner can generate an ordered list $T = ((S_1, h_1, u_1), \ldots (S_n, h_n, u_n)))$ of subsamples of $S$ that are realizable with respect to some $u \in \mathcal{U}$, such that $(h_i, u_i) = \mathrm{MRERM}_{\mathcal{H}}^{\mathcal{U}}(S_i)$ for all $i$ and $u_i \preceq_{\mathcal{H}} u_j$ for all $i \leq j$. Note that we have $u^* \preceq_{\mathcal{H}} u_i$ if and only if the perfect attack oracle returns robust for all sample points in $S_i$ when tested on $h_i$, that is $\mathcal{O}_{u^*}(h_i, x) = \mathrm{robust}$ for all $x$ with $(x, y) \in S_i$ for some $y$. Thus, using a binary search over the list $T$, we can identify the smallest index $\iota$ for which $u^* \preceq_{\mathcal{H}} u_\iota$. Since we have $n \leq 2^m$ and every test in the search employs at most $m$ calls to the perfect attack oracle, this search terminates after at most $m^2$ queries. Finally, note that all subsets $S_i$ from $T$ for which $i \geq \iota$ are realizable with respect to $u^*$. Thus we can choose any largest among $S_\iota, \ldots S_n$ as a maximal realizable subset with respect to the unknown perturbation type $u^*$. $\qquad\square$

In Section B we provide additional guarantees for the agnostic case.

## 4 Beyond totally ordered perturbation classes

We now consider a more general setting where $\mathcal{U}$ is not necessarily totally ordered with respect to $\mathcal{H}$.

### 4.1 Learning with the standard adversarial loss

We will start by showing that $\mathcal{H}, \mathcal{U}$ are learnable with access to a perfect attack oracle if $\mathcal{U}$ is the union of $k$ sub-classes which are each totally ordered with respect to $\mathcal{H}$. This naturally contains the case that $\mathcal{U}$ is finite. Another natural case where this occurs is considering perturbation sets that are balls around domain points of certain radii and with respect to one of a number of possible norms. Consider $\mathcal{X} = \mathbb{R}^d$ and the set

$$\mathcal{U} = \{u_r^p : x \mapsto B_r^p(x) \mid r \in \mathbb{R}, \ p \in 0, 1, 2, \infty\}$$

where we let $B_r^p(x)$ denote the $\ell^p$-norm ball around $x$. Then the above class $\mathcal{U}$ is a union of 4 totally ordered perturbation classes (one for balls of each norm) with respect to any hypothesis class $\mathcal{H}$.

**Theorem 3.** *Let $\mathcal{H}$ be a hypothesis class with $\mathrm{VC}(\mathcal{H}) = d < \infty$, and let $\mathcal{U}$ be a perturbation class, which is a union $\mathcal{U} = \mathcal{U}_1 \cup \mathcal{U}_2 \cup \ldots \cup \mathcal{U}_k$ of $k$ subclasses which are each totally ordered with respect to class $\mathcal{H}$. Then $\mathcal{H}$ is learnable with access to a perfect attack oracle $\mathcal{O}_{u^*}$ in the realizable case with respect to $\mathcal{U}$ with sample complexity $O(\frac{d \cdot 2^d \cdot \log(k/\delta)}{\epsilon} + \frac{\log(k) \cdot \log(1/\epsilon) + \log(1/\delta)}{\epsilon^2})$ and query complexity $O(\left(\frac{\log(k) \cdot \log(1/\epsilon) + \log(1/\delta)}{\epsilon^2}\right)^2).$*

*Proof.* We can generate $k$ hypotheses $h_1, h_2, \ldots, h_k$ that are the output of an $(\epsilon, \delta)$-successful learner for $\mathcal{H}$ with respect to each of the $\mathcal{U}_i$. Since each of the classes $\mathcal{U}_i$ are totally ordered with respect to $\mathcal{H}$, the result of Theorem 1 tells us that $O(\frac{d \cdot 2^d \cdot \log(k/\delta)}{\epsilon})$ sample points suffice to guarantee that, with probability at least $1 - \delta$, we have $\ell^{u^*}(h_i) \leq \epsilon$ for all $i$ with $u^* \in \mathcal{U}_i$. Thus the class $\mathcal{H}_k = \{h_1, h_2, \ldots, h_k\}$ of these $k$ learning outputs has approximation error at most $\epsilon$ with respect to the data generating distribution $P$. Note that, since we have $|\mathcal{H}_k| = k$, this class has also bounded Littlestone dimension at most $\log(k)$. Employing a result (Montasser et al. [2021], Theorem 3) on agnostically learning finite Littlestone classes with access to a perfect attack oracle, an additional $O(\frac{\log(k) \cdot \log(1/\epsilon) + \log(1/\delta)}{\epsilon^2})$ samples and the stated number of queries suffice to output a hypothesis $h$ with $\mathcal{L}_P^{u^*}(h) \leq 3\epsilon$. $\qquad \square$

It is not difficult to see that, even in the realizable case, and when $\mathcal{U}$ is the union of just two totally ordered subclasses, a learner without access to a perfect attack oracle cannot succeed with respect to the standard adversarial loss.

**Observation 2.** *There exists a class $\mathcal{H}$ with $\mathrm{VC}(\mathcal{H}) = 1$ and a perturbation class $\mathcal{U}$ that is the union of two totally ordered subclasses with respect to $\mathcal{H}$ that are not learnable in the realizable case (in the sense of Definition 3) without access to a perfect attack oracle.*

For the proof of this observation we refer the reader to the supplementary material. Now we show that if we remove even more structure from the set of perturbation types $\mathcal{U}$ then learning success becomes impossible even with access to a perfect attack oracle. The following observation will then motivate to investigate learning with respect to the adversarial abstention loss (rather than the standard adversarial loss).

**Observation 3.** *There exists a class $\mathcal{H}$ with $\mathrm{VC}(\mathcal{H}) = 1$ and a perturbation class $\mathcal{U}$ that are not learnable (in the sense of Definition 3), even with access to a perfect attack oracle.*

*Proof.* We consider the same domain, class, and distribution as in the proof of the previous observation: $\mathcal{X} = \mathbb{R}$, $P$ with $P((-3, 1) = P((3, 0)) = 0.5$, and

$$\mathcal{H} = \{h_t : x \mapsto \mathbb{1}\left[x \leq t\right] \mid t \in \mathbb{R}\}.$$

Further, we consider the class $\mathcal{U} = \{u_{a,b} \mid a, b \in \mathbb{R}\}$ of perturbation types $u_{a,b}$, that assign all points $x \leq 0$ to a ball of radius $a$ around $x$ and all points $x > 0$ to a ball of radius $b$. Then, even when promised realizability, a learner cannot distinguish the cases where $u^* = u_{a,b}$ for some values $a, b$ with $a + b \leq 6$ from samples and any finite number of queries to the perfect attack oracle. In particular, if $u^* = u_{a,b}$ for some $a, b$ with $a + b = 6$, predicting with respect to one such pair $a, b$ will induce loss 0.5 for all other such pairs. $\qquad \square$

## 4.2 Learning with the adversarial abstention loss

We will now consider learning with respect to the adversarial abstention loss. We will first show that learning with abstentions can actually help overcome some of the impossibilities shown above. For this, let us revisit the example seen in Observation 3.

**Observation 4.** *There exists a class $\mathcal{H}$ with $\mathrm{VC}(\mathcal{H}) = 1$ and a perturbation class $\mathcal{U}$ that are not learnable (in the sense of Definition 3), even with access to a perfect attack oracle, but that is robustly learnable with abstentions (in the sense of Definition 5) in the realizable case (without access to a perfect attack oracle).*

*Proof sketch.* We consider the example from the proof of Observation 3. Now consider the sub-classes $\mathcal{U}_1 = \{u_{a,0} : a \in \mathbb{R}\}$ and $\mathcal{U}_2 = \{u_{0,b} : b \in \mathbb{R}\}$. Since both $\mathcal{U}_1$ and $\mathcal{U}_2$ are totally ordered we can learn successful hyptheses $h_1$ and $h_2$ respectively. We now define $h : \mathcal{X} \to \{0, 1, \star\}$ by $h(x) = y$ if $h_1(x) = h_2(x) = y$ and $h(x) = \star$ otherwise. This is a successful hypothesis. For more detail, we refer the reader to the appendix. $\qquad\square$

We will now continue by showing that, similar to the case where we have access to a perfect-attack-oracle, a finite union $\mathcal{U} = \bigcup_{i=1}^{k} \mathcal{U}_i$ of totally ordered perturbation sets $\mathcal{U}_i$ is robustly learnable with abstention in the robustly realizable case.

**Theorem 4.** *Let $\mathcal{U} = \bigcup_{i=1}^{k} \mathcal{U}_i$, where every $\mathcal{U}_i$ is totally ordered. Furthermore, let $\mathcal{H}$ have a finite VC dimension. Then $\mathcal{H}$ is robustly learnable with abstention with respect to $\mathcal{U}$ in the robustly realizable case with sample complexity $O(\frac{d \cdot 2^d \cdot k^2 \log(k/\delta)}{\epsilon})$.*

### 4.2.1 Abstention learning with disagreement coefficient

The proof of Theorem 4 shows how bounding the mass of the region on which two learners with small 0/1-loss disagree, yields a robust error guarantee for learning with abstentions. In this section, we will review the definition of the *disagreement coefficient* (Hanneke [2007], Hanneke et al. [2014]), and show that hypothesis classes $\mathcal{H}$ with finite disagreement coefficient are robustly learnable with abstention in the realizable case with respect to every class of perturbation types $\mathcal{U}$.
For a distribution $P_\mathcal{X}$ over $\mathcal{X}$, we let the $P_\mathcal{X}$-induced difference between two hypotheses be denoted by
$$d_{P_\mathcal{X}}(h, h') = \mathbb{P}_{x \sim P_\mathcal{X}}[h(x) \neq h'(x)].$$
The ball in $\mathcal{H}$ around a hypothesis $h$ with radius $r$ is then denoted by
$$B_{P_\mathcal{X}}(h, r) = \{h' \in \mathcal{H} : d_{P_\mathcal{X}}(h, h') \leq r\}.$$

Lastly, let the disagreement region of a set of hypotheses $\mathcal{H}$ be
$$\mathrm{DIS}(\mathcal{H}) = \{x \in \mathcal{X} : \exists h, h' \in \mathcal{H} \text{ with } h(x) \neq h'(x)\}.$$

We can now state the definition of the disagreement coefficient.

**Definition 6** (Hanneke et al. [2014]). *For a distribution $P$ over $\mathcal{X} \times \mathcal{Y}$ and a class $\mathcal{H} \subseteq \{0, 1\}^\mathcal{X}$, let $h^* \in \arg\min_{h' \in \mathcal{H}} \mathcal{L}_P(h')$. The disagreement-coefficient of $\mathcal{H}$ with respect to $P$ is defined by:*
$$\theta_P(\mathcal{H}) = \sup_{r \in (0,1)} \frac{P_\mathcal{X}(\mathrm{DIS}(B(h^*, r)))}{r}.$$

*Furthermore, let the worst-case disagreement coefficient be $\theta(\mathcal{H}) = \sup_{P \in \Delta(\mathcal{X} \times \mathcal{Y})} \theta_P(\mathcal{H})$.*

**Theorem 5.** *Let $\mathcal{H}$ be a hypothesis class with $\theta(\mathcal{H}) < \infty$ and $VC(\mathcal{H}) = d < \infty$. Then the class $\mathcal{H}$ is robustly learnable with abstention in the realizable case with respect to the class of all perturbation types $\mathcal{U}^{\mathrm{all}}$ with sample complexity $O(\frac{d + \log(\frac{1}{\delta})}{(\frac{\epsilon}{\theta(\mathcal{H})})^2})$.*

The proof can be found in Appendix D.6, but the main idea is that for any point $x$, we return a $y \in \{0, 1\}$ if and only if every $h$ with $\mathcal{L}_S^{0/1}(h) = 0$ agrees with $y$, otherwise we abstain. Because of the bound on the disagreement coefficient, one can show that for a large enough $S$, this leads to a hypothesis with low abstention loss.

### 4.2.2 Abstention learning via $\epsilon$-cover

We will now generalize our previous results, by introducing the concept of an $(\epsilon, \mathcal{H})$-cover.

**Definition 7** ($(\epsilon, \mathcal{H})$-cover of $\mathcal{U}$). *Let $\mathcal{H} \subseteq \{0, 1\}^\mathcal{X}$ be a hypothesis class, $P$ a distribution over $\mathcal{X} \times \mathcal{Y}$ and $\mathcal{U}$ a class of perturbation types. A set of hypotheses $\mathcal{H}'$ is a $(\epsilon, \mathcal{H})$-cover for $\mathcal{U}$ with respect to $P$, if for every $u \in \mathcal{U}$, there exists an $h' \in \mathcal{H}'$ such that $\mathcal{L}_P^u(h') < \inf_{h \in \mathcal{H}} \mathcal{L}_P^u(h) + \epsilon$.*

We will now provide the definition of a successful cover learner.

**Definition 8.** *Let $\mathcal{H} \subseteq \{0,1\}^{\mathcal{X}}$ be a hypothesis class, and $\mathcal{U}$ be a class of perturbation types. We say a function $\mathcal{A}_{cover} : (\mathcal{X} \times \mathcal{Y})^* \to 2^{\{0,1\}^{\mathcal{X}}}$ that maps a sample $S$ to a hypothesis class $\hat{\mathcal{H}} \subset \mathcal{F}$ is a successful finite-disagreement-cover learner for $(\mathcal{H}, \mathcal{U})$, if there is a function $m_{(\mathcal{H},\mathcal{U}),cover} : (0,1)^3 \to \mathbb{N}$, such that for every $\epsilon, \eta, \delta \in (0,1)$, every $m \geq m_{(\mathcal{H},\mathcal{U}),cover}(\epsilon,\eta,\delta)$ and every distribution $P$ over $\mathcal{X} \times \{0,1\}$ with probability $1 - \delta$ over $S \sim P^m$ both of the following statements hold:*

- *$\mathcal{A}(S)$ is an $(\epsilon, \mathcal{H})$-cover for $\mathcal{U}$ with respect to $P$*

- *$P_{\mathcal{X}}(\mathrm{DIS}(\mathcal{A}(S)) \leq \eta$ .*

We note that any successful cover learner, that is guaranteed to output a finite set is a successful finite-disagreement cover learner. The case, where $\mathcal{U}$ is a union of $k$ totally ordered perturbation type classes $\mathcal{U}_i$, we have seen before, can be interpreted as first learning a cover $\hat{H}$ of size $k$, where each of the element of the union "covers" one of the classes. The case in which $\mathcal{H}$ has a finite disagreement coefficient can furthermore be seen as a case where the set of all $\mathrm{ERM}_{\mathrm{H}}$ hypotheses constitute an $(\epsilon, \mathcal{H})$-cover $\mathcal{H}'$ for every class of perturbation types $\mathcal{U}$. The mass of the disagreement region of $\mathcal{H}$ is then bounded by the disagreement coefficient. We now state our result generalizing these observations.

**Theorem 6.** *A class $\mathcal{H}$ is robustly learnable with abstentions with respect to $\mathcal{U}$ in the robustly realizable case, if there is a successful finite-disagreement-cover learner for $(\mathcal{H}, \mathcal{U})$. Furthermore, the sample complexity of robustly learning with abstentions is then bounded by $m_{\mathcal{H},\mathcal{U}}(\epsilon,\delta) \leq m_{\mathcal{H},\mathcal{U},cover}(\epsilon/2, \epsilon/2, \delta)$.*

Inspired by considerations of adaptive robustness radii [Chowdhury and Urner, 2022, Bhattacharjee and Chaudhuri, 2021], in Section C of the appendix we discuss a setting of abstention learning in which we allow the size of the perturbation sets to vary locally between different regions of the domain. We provide a learning guarantee for this setting that is not covered by our previous results.

# 5   Conclusion

In this work we relaxed the assumption that we know exactly the perturbation sets that are to be used by an adversary. Our model provides an interpolation between knowing the perturbation sets exactly and not knowing them at all. Many of our results rely on realizability. We have also shown that without realizability, learning is not possible unless the learner receives other feedback, such as access to a perfect attack oracle. It might be interesting to consider some milder relaxations of realizability. For example, instead of saying $\mathrm{opt}_P^{u^*}(\mathcal{H}) = 0$, what if we only know that the adversarial Bayes loss is zero for some $u^* \in \mathcal{U}$?

While our work focuses on statistical aspects, it will also be interesting to understand the kinds of computation required for our results. For example, we rely on the existence of an MRERM oracle. When are they computationally feasible? Do we require accurate MRERM oracles or can approximate (and hence potentially more tractable) versions also be used for the learning task?

Another question is to find a dimension that characterizes learning in various settings. This was only recently resolved for adversarially robust learning (for the case when $u$ is known) Montasser et al. [2022], however, the dimension proposed in the paper is not easy to define. On the other hand, in the setting when the only information we can obtain about $u$ is through a perfect attack oracle, the Littlestone dimension of $\mathcal{H}$ has been shown to characterize adversarially robust learning Montasser et al. [2021]. It will be interesting to see if such a simple dimension can be obtained for the settings considered in our paper.

### Acknowledgements

Ruth Urner is also a faculty affiliate member at Toronto's Vector institute, and acknowledges research support through an NSERC discovery grant. Tosca Lechner is supported by a Vector Research Grant and a Waterloo Apple PhD Fellowship in Data Science and Machine Learning.

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

## A  Maximally Robust Empirical Risk Minimization

Recall the definition of the empirical risk minimization oracles:

$$\mathrm{ERM}_{\mathcal{H}} : S \mapsto h_S \in \mathrm{argmin}_{h \in \mathcal{H}} \mathcal{L}_S^{0/1}(h)$$
$$\mathrm{RERM}_{\mathcal{H}}^{\mathrm{u}} : S \mapsto h_S^u \in \mathrm{argmin}_{h \in \mathcal{H}} \mathcal{L}_S^u(h)$$
$$\mathrm{MRERM}_{\mathcal{H}}^{\mathcal{U}} : S \mapsto \begin{cases} (h_S^{u_S^*}, u_S^*) \text{ with } h_S^{u_S^*} \in \mathrm{argmin}_{h \in \mathcal{H}} \mathcal{L}_S^{u_S^*}(h), \\ \qquad \text{and } u_S^* \text{ is } \preceq_{\mathcal{H}} \text{ maximal in the set } \{u \in \mathcal{U} \mid \min_{h \in \mathcal{H}} \mathcal{L}_S^u(h) = 0\} \\ \text{error if } \mathcal{L}_S^u(h) > 0 \text{ for all } u \in \mathcal{U} \text{ and } h \in \mathcal{H} \end{cases}$$

While there always exists functions $h_S$ and $h_S^u$ that are solutions to the queries in the first two oracles, the last one requires a bit more care.

Let $S$ be some labeled dataset and assume that $\mathcal{H}$ and $\mathcal{U}$ are hypothesis and perturbation type classes such that $S$ is $\mathcal{H}$-realizable for the true perturbation type $u^* \in \mathcal{U}$. We can then define a maximal perturbation type $u_S^*$ for $S$, $\mathcal{H}$ and $\mathcal{U}$ as follows:

$$u_S^*(x) = \bigcup_{u \in \mathcal{U} \text{ such that } \min_{h \in \mathcal{H}} \mathcal{L}_S^u(h) = 0} u(x).$$

This defines a perturbation type $u_S^*(x)$, which is not necessarily in $\mathcal{U}$ but does satisfy $u^* \preceq_{\mathcal{H}} u_S^*$, which suffices for our purposes.

If, given $u_S^*$, the set $\mathrm{argmin}_{h \in \mathcal{H}} \mathcal{L}_S^{u_S^*}(h)$ is non-empty, the oracle can return any function in this set as $h_S^{u_S^*}$. We will now argue that we can define a "closure" $\bar{\mathcal{H}}$ of the hypothesis class $\mathcal{H}$, which has the same VC-dimension as $\mathcal{H}$ and such that $h_S^{u_S^*} \in \bar{\mathcal{H}}$.

We here show how such an embedding of $\mathcal{H}$ into the closure class $\bar{\mathcal{H}}$ can be constructed in the case of a countable domain $\mathcal{X}$ and a perturbation class $\mathcal{U}$ that is *separable*, in the sense that for every $\mathcal{U}' \subset \mathcal{U}$ there exists a countable subset $\mathcal{U}'' \subset \mathcal{U}'$, such that for every $x \in \mathcal{X} : \bigcup_{u \in \mathcal{U}'} u(x) = \bigcup_{u \in \mathcal{U}''} u(x)$. This is the case for a wide range of perturbation types, in particular in the case for which the sets $u(x)$ are always open balls with respect to some $\ell_p$-norm in $\mathbb{R}^d$. In the general case (where $\mathcal{X}$ may be uncountable and $\mathcal{U}$ not separable in the above sense) recent work has shown that the existence of a closure class with identical VC-dimension is still guaranteed [Lechner et al., 2023]. The more general construction there employs the set-theoretic concept of *filters*. It is shown in the context of strategic classification, but the technique is readily available to adversarial perturbation sets. We here illustrate the intuition behind that abstract construction by focusing on countable and separable case.

We now assume that $\mathcal{X}$ is countable and let $(x_i)_{i \in \mathbb{N}}$ be an enumeration of the elements of $\mathcal{X}$. The separability assumption on the perturbation class $\mathcal{U}$ implies that for all $S$ there is a countable subset $\mathcal{U}^c \subseteq \mathcal{U}$ such that for all $x \in \mathcal{X}$ we have

$$u_S^*(x) = \bigcup_{u \in \mathcal{U}^c \text{ such that } \min_{h \in \mathcal{H}} \mathcal{L}_S^u(h) = 0} u(x).$$

For a given sample $S$, let $(u_i)_{i \in \mathbb{N}}$ be an enumeration of the set

$$\{u \in \mathcal{U}^c \mid \min_{h \in \mathcal{H}} \mathcal{L}_S^u(h) = 0\}$$

and let $(h_i)_{i \in \mathbb{N}}$ be an enumeration of corresponding empirical risk minimizing hypotheses from $\mathcal{H}$:

$$h_i = \mathrm{RERM}_{\mathcal{H}}^{\mathrm{u_i}}(S).$$

We will now define the function $h_S^*$ by successively choosing subsequences from the sequence $(h_i)_{i \in \mathbb{N}}$ as follows. We set $F_0 = (h_i)_{i \in \mathbb{N}}$ and $J_0 = \mathbb{N}$. Then, for every $i \in \mathbb{N}$, we define a set of indices $J_i$ such that

- **Case 1:** There exists a $k \in J_{i-1}$ such that $h_j(x_i) = h_k(x_i)$ for all $j \geq k$, $j \in J_{i-1}$. Then we define:
$$J_i = \{j \in J_{i-1} \mid h_j(x_i) = h_k(x_i)\}$$

- **Case 2:** Both labels are assigned infinitely often to $x_i$ by the sequence $(h_j)_{j \in J_{i-1}}$. Then we define

$$J_i = \{j \in J_{i-1} \mid h_j(x_i) = 1\}$$

and we define subsequence $F_i = (h_j)_{j \in J_i}$. Note that, by construction, in each resulting sequence $F_i$ the first $i$ points $x_1, x_2, \ldots, x_i$ receive the same label by all functions in the sequence. Now, by choosing indices to select the $i$-th function from each sequence $F_i$ (by choosing the first index of $J_1$, the second index in $J_2$, the third in $J_3$ and so on), and re-identifying these indices with the natural numbers, we obtain yet another subsequence $F = (f_i)_{i \in \mathbb{N}}$. In this sequence we now have for every $i \in \mathbb{N}$ and every $k \geq i$

$$f_k(x_i) = f_i(x_i).$$

That is, this sequence of functions converges pointwise and we can use the above equation define the function $h_S^*$ as follows:

$$h_S^{u_S^*}(x_i) = f_i(x_i).$$

It is not difficult to see that the so defined function $h_S^{u_S^*}$ satisfied $\mathcal{L}_{u_S^*}^S(h_S^{u_S^*}) = 0$. Let let $\bar{\mathcal{H}}$ be defined by adding these function $h_S^{u_S^*}$ to $\mathcal{H}$ for all possible $S$.

$$\bar{\mathcal{H}} = \mathcal{H} \cup \{h_S^{u_S^*} \in \{0,1\}^{\mathcal{X}} \mid S \in (\mathcal{X} \times \mathcal{Y})^*\}$$

We observe that every behavior that $\bar{\mathcal{H}}$ exhibits on a finite subset $C \subseteq \mathcal{X}$, is already witnessed by a function from $\mathcal{H}$. Thus we get $\mathrm{VC}(\bar{\mathcal{H}}) = \mathrm{VC}(\mathcal{H})$, and we can always let $\mathrm{MRERM}_{\mathcal{H}}^{\mathcal{U}}$ return a function from $\bar{\mathcal{H}}$.

## B  Agnostic learning for finite unions of total orders

**Theorem 7.** *Let $\mathcal{H}$ be a hypothesis class with $\mathrm{VC}(\mathcal{H}) = d < \infty$, and let $\mathcal{U}$ be a perturbation class, which is a union $\mathcal{U} = \mathcal{U}_1 \cup \mathcal{U}_2 \cup \ldots \cup \mathcal{U}_k$ of $k$ subclasses which are each totally ordered with respect to class $\mathcal{H}$. Then $\mathcal{H}$ is agnostically learnable with access to a perfect attack oracle $\mathcal{O}_{u^*}$ with respect to $\mathcal{U}$ with sample complexity $m_1(\epsilon, \delta) + m_2(\epsilon, \delta)$, and query complexity $n_1(\epsilon, \delta) + n_2(\epsilon, \delta)$ where:*

- $m_1(\epsilon, \delta) = O\left(\frac{d \cdot 2^d \cdot \log(k/\delta)}{\epsilon^2}\right)$

- $m_2(\epsilon, \delta)$ *is the sample complexity of agnostically learning a hypothesis class of size $k$ wrt* $\mathcal{U}^{\mathrm{all}}$.

- $n_1(\epsilon, \delta) = k \cdot (m_1(\epsilon, \delta))^2$

- $n_2(\epsilon, \delta)$ *is the query complexity of agnostically learning a hypothesis class of size $k$ wrt $\mathcal{U}^{\mathrm{all}}$*

*Proof.* We can generate $k$ hypotheses $h_1, h_2, \ldots, h_k$ that are the output of an $(\epsilon, \delta)$-successful learner for $\mathcal{H}$ with respect to each of the $\mathcal{U}_i$. Since each of the classes $\mathcal{U}_i$ are totally ordered with respect to $\mathcal{H}$, the result of Theorem 2 (combined with a union bound) tells us that $m_1(\epsilon, \delta)$ sample points and $n_1(\epsilon, \delta)$ queries suffice to guarantee that, with probability at least $1 - \delta$, we have $\ell^{u^*}(h_i) \leq \mathrm{opt}_P^{u^*}(\mathcal{H}) + \epsilon$ for all $i$ with $u^* \in \mathcal{U}_i$. Thus the class $\mathcal{H}_k = \{h_1, h_2, \ldots, h_k\}$ of these $k$ learning outputs has approximation error at most $\mathrm{opt}_P^{u^*}(\mathcal{H}) + \epsilon$ with respect to the data generating distribution $P$. Note that, since we have $|\mathcal{H}_k| = k$, this class has bounded Littlestone dimension at most $\log(k)$, VC-dimension at most $\log(k)$, and dual VC-dimension at most $k$. Employing a result (Montasser et al. [2021], Theorem 4) on agnostically learning finite VC classes with access to a perfect attack oracle, we get that both $m_2(\epsilon, \delta)$ and $n_2(\epsilon, \delta)$ are finite and $\mathcal{L}_P^{u^*}(h) \leq \mathrm{opt}_P^{u^*}(\mathcal{H}) + 2\epsilon$.  $\square$

## C  $\epsilon$-cover Learning

### C.1  Abstention learning for regions with different radii

Let us now explore the case, where we allow our perturbation types to consist of perturbation sets whose size varies throughout the domain. While a lot of work assumes that the size of the radius

is fixed throughout the domain, it is not a prior clear that the distances in the domain correspond perfectly to what humans find perceptible. This was noted in prior works [Bhattacharjee and Chaudhuri, 2021, Chowdhury and Urner, 2022] and motivated their adaptive adversarial loss. We explore such variation of local radii for the setting of uncertain perturbation sets. We have seen examples of this in Observation 3 and Observation 4. We will now generalize these perturbation types. Let $\mathcal{X} = \mathbb{R}$. For a function $f : \mathcal{X} \to [k]$, let the perturbation type $u_{f,a_1,\dots,a_k}$ defined by:

$$u_{f,a_1,\dots,a_k}(x) = B_{a_{f(x)}}(x).$$

That is, we have $k$ different regions in the domain, which are determined by $f$. Furthermore, we assume that each perturbation set within a region $i$ is a ball with radius of a fixed size $a_i$, while radii can vary between regions. For a fixed function $f$, let us now consider

$$\mathcal{U}_f = \{u_{f,a_1,\dots,a_k} : a_1, \dots, a_k \in \mathbb{R}\}.$$

Let $\mathrm{VC}(\mathcal{H}) \leq \mathrm{VC}(\mathcal{H} \times \mathcal{U})_\ell \leq d$. We note that in general, that is excluding particular hypothesis sets $\mathcal{H}$ and function $f$, this set of perturbation sets cannot be expressed as a finite union of totally ordered sets.

For every $i \in [k]$ consider the classes $\mathcal{U}_{f,i} = \{u_{f,0,\dots,0,a_i,0,\dots,0} : a_i \in \mathbb{R}\}$. Let $P$ be a realizable distribution with respect to $\mathcal{H}$ and $u^* \in \mathcal{U}$. Let $m(\epsilon, \delta) = C \frac{d + \log(\frac{1}{\delta})}{\epsilon^2}$, where $C$ is the constant we require to get a uniform-convergence guarantee (which exists due to the corresponding VC-dimensions being finite, see Shalev-Shwartz and Ben-David [2014]), and let $m \geq m(\frac{\epsilon}{k(2k^2+1)}, \frac{\delta}{k})$. Now for a sample $S$ and any $i$, let $(h_i, u_i) = \mathrm{MRERM}_{\mathcal{U}_{f,i}}^{\mathcal{H}}(S)$. We denote by $a_i^{max}$ the maximum realizable radius for component $i$, i.e. $h_{0,\dots,0,a_i^{max},0,\dots,0} = h_i$. Now let $u^* = u_{f,a_1^*,\dots,a_k^*}$ be the ground true perturbation set. Then for every $i$, $a_i^* \leq a_i^{max}$. Let $u^{max} = u_{f,a_1^{max},\dots,a_k^{max}}$. We note that for every $x \in \mathcal{X}, y \in \mathcal{Y}$ and $h \in \mathcal{F}$, we have $\ell^{u^{max}}(h, x, y) \leq \max_{i \in [k]} \ell^{u_i}(h, x, y)$. Now let

$$h(x) = \begin{cases} y & \text{if for all } i : h_i(x) = y \\ \star & \text{otherwise.} \end{cases}$$

We note that by the PAC-learning guarantee with respect to $0/1$-loss, with probability $1 - \delta$, for every $i, j \in [k]$ we have a pairwise disagreement between $h_i$ and $h_j$ of at most $\frac{2\epsilon}{(2k^2+1)k}$. Thus the disagreement-region of $\{h_i : i \in [k]\}$ has mass at most $\frac{k^2\epsilon}{k(2k^2+1)}$. Thus, with probability $1 - \delta$ over the sample generation the abstention loss of $h$ we have

$$\mathcal{L}_P^{u^*,\mathrm{abst}}(h) \leq \mathcal{L}_P^{u^{max},\mathrm{abst}}(h) \leq \sum_{i=1}^{k} \mathcal{L}_P^{u^i,\mathrm{abst}}(h) \leq \sum_{i=1}^{k} (\mathcal{L}_P^{u^i}(h_i) + P_{\mathcal{X}}(\{x : h(x) = \star\}))$$

$$\leq k(\frac{\epsilon}{k(2k^2+1)} + \frac{k^2\epsilon}{k(2k^2+1)}) = \epsilon.$$

Note, that we can thus view $\{h\}$ as an $(\mathcal{H}, \epsilon)$-cover of $\mathcal{U}$ with respect to $P$ and adversarial *abstention* loss.

## D  Proofs

### D.1  Proof of Theorem 1

*Proof.* For this result we adapt a compression based argument for the case of fixed perturbation type [Montasser et al., 2019] to the case of a class $\mathcal{U}$ of perturbation types. A similar adaptation has recently been shown for the setting of strategic classification [Lechner et al., 2023]. The key difference to the original compression scheme is that we use the $\mathrm{MRERM}_{\mathcal{H}}^{\mathcal{U}}$ oracle for the weak learners in each round of boosting, rather than the simple $\mathrm{RERM}_{\mathcal{H}}^{\mathrm{u}}$ oracle as is done in the original publication [Montasser et al., 2019]. Importantly, the maximal ERM paradigm is well defined on all subsets of an original given data sample, and thus each required weak hypothesis can be encoded with a finite number of samples. Finally, since the maximal perturbation type for any subsample is always at least as large as the true perturbation type (under the realizablity assumption), using $\mathrm{MRERM}_{\mathcal{H}}^{\mathcal{U}}$ for each boosting step we obtain the required guarantee for the true perturbation type (without the learner/compressor requiring knowledge of the true type). We thus obtain the same compression size and implied sample complexity in the case of a totally ordered class of perturbation types as for a fixed perturbation type. $\qquad\square$

## D.2 Proof of Observation 1

*Proof.* Consider a finite domain $\mathcal{X} = \{x_1, x_2, x_3\}$ and distribution $P$ over $\mathcal{X} \times \{0,1\}$ with $P((x_1, 1)) = 1/4$, $P((x_3, 0)) = 3/4$ and $P((x, y)) = 0$ for all $(x, y) \notin \{(x_1, 1), (x_3, 0)\}$. We define a class $\mathcal{U} = \{u_1, u_2\}$ containing two perturbation types $u_1$ and $u_2$ defined as follows:

$$u_1(x_1) = \{x_1, x_2\}, \ u_1(x_2) = \{x_2\}, \ u_1(x_3) = \{x_3\}$$

and

$$u_2(x_1) = \{x_1, x_2\}, \ u_2(x_2) = \{x_2\}, \ u_2(x_3) = \{x_2, x_3\}$$

We let $h_{y_1 y_2 y_3}$ denote the function that labels $x_i$ with $y_i$ on this domain (there are only 8 different non-abstaining predictors over $\mathcal{X}$), and let $H = \{h_{110}, h_{100}\}$ be a hypothesis class containing only two of these predictors. Clearly $\text{VC}(\mathcal{H}) = 1$ and $u_1 \preceq_{\mathcal{H}} = u_2$. Note that

$$\text{opt}_P^{u_1}(\mathcal{H}) = \mathcal{L}_P^{u_1}(h_{110}) = 0$$

while $\mathcal{L}_P^{u_1}(h_{y_1 y_2 y_3}) \geq 1/4$ for all predictors $h_{y_1 y_2 y_3} \neq h_{110}$. We further note that

$$\text{opt}_P^{u_2}(\mathcal{H}) = \mathcal{L}_P^{u_2}(h_{100}) = \mathcal{L}_P^{u_2}(h_{000}) = 1/4$$

while $\mathcal{L}_P^{u_2}(h_{y_1 y_2 y_3}) \geq 3/4$ for all predictors $h_{y_1 y_2 y_3} \notin \{h_{000}, h_{100}\}$. Samples from $P$ will not allow distinguishing whether the true perturbation type is $u_1$ or $u_2$ (since we are not imposing a realizability assumption here). Thus, no learner will be able to output predictors that are (close to) optimal for all possible perturbation types in $\mathcal{U}$. $\square$

## D.3 Proof of Observation 2

*Proof.* We consider $\mathcal{X} = \mathbb{R}$ and $\mathcal{H}$ to be the class of threshold classifiers:

$$\mathcal{H} = \{h_t : x \mapsto \mathbb{1}\,[x \leq t] \mid t \in \mathbb{R}\}.$$

Further consider $\mathcal{U} = \mathcal{U}_- \cup \mathcal{U}_+$, where $\mathcal{U}_-$ consists of perturbation types $u_r^-$ that assign each $x$ to a ball (interval) of radius $r$ around $x$ if $x \leq 0$ and a ball of radius $r/2$ around $x$ if $x > 0$, and $\mathcal{U}_+$ similarly consists of perturbation types $u_r^+$ that assign each $x$ to a ball (interval) of radius $r$ around $x$ if $x > 0$ and a ball of radius $r/2$ around $x$ if $x \leq 0$. Both $\mathcal{U}_-$ and $\mathcal{U}_+$ are totally ordered (with respect to any hypothesis class) but their union $\mathcal{U}$ is not totally ordered with respect to $\mathcal{H}$. Consider a distribution $P$ with $P((-3, 1) = P((3, 0)) = 0.5$. Then, even when promised realizability, a learner cannot distinguish the cases $u^* = u_4^-$ and $u^* = u_4^+$ from samples; and performing well with respect to one of these perturbation types will induce loss $0.5$ with respect to the other. $\square$

## D.4 Proof of Observation 4

*Proof.* We consider the example from Observation 3 Now consider the classes $\mathcal{U}_1 = \{u_{a,0} : a \in \mathbb{R}\}$ and $\mathcal{U}_2 = \{u_{0,b} \in \mathbb{R}\}$. Since both $\mathcal{U}_1$ and $\mathcal{U}_2$ are totally ordered there are successful robust learners $\mathcal{A}_1$ for $\mathcal{H}$ with respect to $\mathcal{U}_1$ and $\mathcal{A}_2$ with respect to $\mathcal{U}_2$. We run both learners on a sample with a size to guarantee $(\epsilon/2, \delta/2)$-success in both cases. We denote the resulting hypotheses with $h_1$ and $h_2$ respectively. We now define $h : \mathcal{X} \to \{0, 1, \star\}$ by $h(x) = y$ if $h_1(x) = h_2(x) = y$ and $h(x) = \star$ otherwise.

We now argue that with probability $1 - \delta$, $h$ has adversarial abstention loss less than $\epsilon$. Let $u^* = u_{a^*, b^*} \in \mathcal{U}$ be the ground-truth perturbation type. Now let $P$ be any $\ell^{u_{a^*, b^*}}$-realizable distribution. Let $(\cdot, u_{a_1, 0}) = \text{MRERM}_{\mathcal{H}}^{\mathcal{U}_1}(S)$ and $(\cdot, u_{0, b_2}) = \text{MRERM}_{\mathcal{H}}^{\mathcal{U}_2}(S)$. With probability 1, we have both $a_1 > a^*$ and $b_2 > b^*$. Further, we note every $x \in \mathcal{X}$ and every $u_{a,b} \in \mathcal{U}$, $\ell^{u_{a,b}}(h, x, y)$ is either determined by $a, h$ and $y$ or $b, h$ and $y$, not both, since either $x \leq 0$ or $x > 0$. This means, that for every $x \in \mathcal{X}$, $\ell^{u_{a_1, b_2}}(h, x, y) \leq \max\{\ell^{u_{a_1, 0}}(h, x, y), \ell^{u_{0, b_2}}(h, x, y)\}$.

Combining these observations, we get

$$\mathcal{L}_P^{u^*, \text{abst}}(h) \leq \mathcal{L}_P^{u_{a_1, b_2}, \text{abst}}(h) \leq \mathcal{L}_P^{u_{a_1, 0}, \text{abst}}(h) + \mathcal{L}_P^{u_{0, b_2}, \text{abst}}(h) \leq \frac{\epsilon}{2} + \frac{\epsilon}{2} = \epsilon.$$

$\square$

## D.5 Proof of Theorem 4

*Proof.* We assume robust realizability with respect to $\mathcal{U}$. In particular this means that we are in the non-robustly realizable case. Thus for any $\mathcal{U}_i$, we know that the set $\mathcal{U}_i' = \mathcal{U}_i \cup \{\{\{x\} : x \in \mathcal{X}\}\}$ is totally ordered and that it contains some $u_i \in \mathcal{U}_i'$ such that $\mathcal{H}$ is robustly realizable with respect to the ground truth distribution $P$ and $u_i$. We know from Theorem 1, that for every $i$, there is a successful maximally robust learner $\mathcal{A}_i$ for $\mathcal{H}$ with respect to $\mathcal{U}_i'$ in the realizable case, since $\mathcal{U}_i'$ is totally ordered.

Now let $\mathcal{A}$ be defined by

$$A(S)(x) = \begin{cases} 1 & \text{iff for all } i \in [k] \text{ we have } \mathcal{A}_i(S)(x) = 1 \\ 0 & \text{iff for all } i \in [k] \text{ we have } \mathcal{A}_i(S)(x) = 0 \\ \star & \text{otherwise .} \end{cases}$$

Furthermore, let $(h, u_i(S)) = \text{MRERM}_{\mathcal{H}}^{\mathcal{U}_i}(S)$. Now let $u^* \in \mathcal{U}$ be the ground-truth perturbation type. We know that there is $i^* \in [k]$, such that $u^* \preceq u_{i^*}$. Let $P$ be a distribution with $\text{opt}_P^{u^*}(\mathcal{H}) = 0$. Now if $S \sim P^m$ with $m \geq \max_{i \in [k]} m(\frac{\epsilon}{2k^2+1}, \frac{\delta}{k})$, then with probability $1 - \delta$, for every $i \in [k]$, $\mathcal{L}_P^{u_i}(\mathcal{A}_i(S)) \leq \frac{\epsilon}{2k^2+1}$ and therefore, $\mathcal{L}_P^{0/1}(\mathcal{A}_i(S)) \leq \frac{\epsilon}{2k^2+1}$. Thus, for any two $i, j \in [k]$, we have

$$P_{\mathcal{X}}(\{x \in X : \mathcal{A}_i(S)(x) \neq \mathcal{A}_j(S)(x)\}) \leq \frac{2\epsilon}{2k^2+1}.$$

Thus,

$$\begin{aligned} \mathcal{L}_P^{u^*,\text{abst}}(\mathcal{A}(S)) &\leq \mathcal{L}_P^{u^*,\text{abst}}(\mathcal{A}_{i^*}(S)) + P_{\mathcal{X}}(\{x \in \mathcal{X} : A(S)(x) = \star\}) \\ &\leq \mathcal{L}_P^{u_{i^*},\text{abst}}(\mathcal{A}_{i^*}(S)) + P_{\mathcal{X}}(\{x \in X : \exists i, j, \mathcal{A}_i(S)(x) \neq \mathcal{A}_j(S)(x)\}) \\ &\leq \frac{\epsilon}{2k^2+1} + \frac{2k^2\epsilon}{2k^2+1} = \epsilon. \end{aligned}$$

$\square$

## D.6 Proof of Theorem 5

*Proof.* Let $u^* \in \mathcal{U}^{\text{all}}$ be any perturbation set and $P$ any distribution with $\text{opt}_P^{u^*}(\mathcal{H}) = 0$. Furthermore, let $m \geq m_{\mathcal{H}}(\epsilon, \delta)$ where $m_H$ is the sample complexity function in $O(\frac{d+\log(\frac{1}{\delta})}{(\frac{\epsilon}{\theta(\mathcal{H})})^2})$. Let $S \sim P^m$. Let $\hat{\mathcal{H}} = \{h \in \mathcal{H} : \mathcal{L}_S^{0/1}(h) = 0\}$. Now we define our output hypothesis $\hat{h}$ by

$$\hat{h} = \begin{cases} 1 & \text{if for every } h \in \hat{\mathcal{H}} : h(x) = 1 \\ 0 & \text{if for every } h \in \hat{\mathcal{H}} : h(x) = 0 \\ \star & \text{otherwise.} \end{cases}$$

We note that there is some $h^* \in \hat{\mathcal{H}}$ with $\mathcal{L}_P^{u^*}(h^*) = \mathcal{L}_P^{0/1}(h^*) = 0$. Since every $h \in \hat{H}$ is an $\text{ERM}_H$ hypothesis and $\mathcal{H}$ has VC dimension $d$, the sample of ERMs gives us

$$\begin{aligned} \mathcal{L}_P^{u^*,\text{abst}}(\hat{h}) &\leq \mathcal{L}_P^{u^*,\text{abst}}(h^*) + P_{\mathcal{X}}(\{x \in \mathcal{X} : h^*(x) \neq \hat{h}(x)\}) \\ &\leq \mathcal{L}_P^{u^*}(h^*) + P_{\mathcal{X}}(\{x \in \mathcal{X} : \hat{h}(x) = \star\}) \\ 0 &+ P_{\mathcal{X}}(\text{DIS}(B(h^*, \frac{\epsilon}{\theta(\mathcal{H})}))) = \epsilon. \end{aligned}$$

$\square$

## D.7 Proof of Theorem 6

*Proof.* Let $m \geq m_{\mathcal{U}, \mathcal{H}, cover}(\epsilon/2, \epsilon/2, \delta)$ and $S \sim P^m$. And let $u^* \in \mathcal{U}$ the ground-truth perturbation set. We define the abstention learner as:

$$\mathcal{A}(S)(x) = \begin{cases} 1 & \text{if for all } h \in \mathcal{A}_{cover}(S) : h(x) = 1 \\ 0 & \text{if for all } h \in \mathcal{A}_{cover}(S) : h(x) = 0 \\ \star & \text{otherwise} \end{cases}$$

Then with probability $1 - \delta$, $P_{\mathcal{X}}(\text{DIS}(\mathcal{A}_{cover}(S))) \leq \frac{\epsilon}{2}$ and $\exists h \in \mathcal{A}_{cover}(S)$ with

$$\mathcal{L}_P^{u^*}(h) \leq \min_{h' \in \mathcal{H}} \mathcal{L}_P^{u^*}(h') + \epsilon/2.$$

Since for every $x \in \mathcal{X}$ either $A(S)(x) = h(x)$ or $x \in \mathcal{A}_{cover}(S)$. Thus,

$$\mathcal{L}_P^{u^*,\text{abst}}(A(S)) \leq \mathcal{L}_P^{u^*,\text{abst}}(h) + P_{\mathcal{X}}(\text{DIS}(\mathcal{A}_{cover}(S))) \leq \min_{h' \in \mathcal{H}} \mathcal{L}_P^{u^*,\text{abst}}(h') + \epsilon.$$

$\square$

