# OpenReview forum: "Adversarially Robust Learning with Uncertain Perturbation Sets"
_NeurIPS.cc/2023/Conference — NeurIPS 2023 poster_

### Official Review · Reviewer_wkxJ · 2023-06-28

**Soundness:** 3 good
**Presentation:** 2 fair
**Contribution:** 3 good
**Rating:** 6
**Confidence:** 3

**Summary:**

The paper studies adversarially robust learning with uncertain perturbation sets, where the set of points to which an adversary can perturb any test input is random, in contrast to previously studied settings of fixed known or unknown perturbation sets and unknown perturbation sets. The perturbation set is assumed to be coming from some fixed perturbation class.

Given a perfect attack oracle which certifies robustness or gives a successful attack, learnability is shown for concept classes with finite VC dimension with finite bounds on sample complexity and query complexity to the oracle, with some assumptions on U which are satisfied by commonly studied perturbation classes e.g. finite union of totally ordered perturbation sets which includes $L_p$ balls for a finite collection of $p$ values.

When the classifier is equipped with abstention, learnability is possible under weaker assumptions e.g. finite disagreement coefficient of H wrt the data distribution. The authors further propose new notion of finite disagreement cover that depends on the perturbation class U, and reduce robust learning under abstentions to successful learning of a finite disagreement cover in the sense of their proposed notion.

**Strengths:**

- The work considers removing a common and unrealistic restriction in prior theoretical work that the adversary uses a fixed perturbation set.
- The authors give examples demonstrating power of abstention and the difficulty of agnostic learning under their definitions.
- Proofs or proof sketches of several major results are included and discussed.
- Comparison to prior works and techniques e.g. Montasser et al. 2021 helps clarify the novelty and technical contributions of the present work.

**Weaknesses:**

- Realizability: Realizability of perturbation set seems like a strong assumption on the adversary and should be discussed. No positive results are given in the main body in the agnostic case, which could be a limitation of the model under study.
- Some results are not adequately discussed, e.g. Theorem 5 and last paragraph on Pg 9.

**Questions:**

Questions:
- Are there any practical examples or approximations of the oracle considered?
- What is the sample complexity in Theorem 4?
- Are the results applicable when the adversary uses a perturbation set outside of U?

Suggestions:
- It is not clear initially (e.g. lines 42-48) that the realizability being talked about is wrt U and not H.
- Typos: lines 113, 358, 364.
- Consider stating positive results in the agnostic case when learning with abstention in the main body (lines 408-413).

**Limitations:**

I do not anticipate potential negative impacts as the work is primarily theoretical, but encourage the authors to summarize limitations and further questions in a conclusion section e.g. lack of positive results in the agnostic case, tightness of sample complexity bounds, etc.

---

> ### Author Rebuttal · Authors · 2023-08-10
>
> > What is the sample complexity in Theorem 4?
>
> Thanks for pointing out the missing bound! We will fix this in the final version. Currently, it’s in the proof in the supplementary material.
>
> The bound is $O(\frac{d\cdot 2^d k^2 \log (k/\delta)}{\epsilon})$, i.e., we get a quadratic dependence on $k$ in addition to the bound from Theorem 1.
>
> > Are the results applicable when the adversary uses a perturbation set outside of U?
>
> If we know nothing about the perturbation type used by the adversary, then this is equivalent to assuming U to be the set of all perturbation types (except for agnostic learning with respect to U which we address below). Thus the known results for  “unknown perturbation sets” (which we reference, line 460) apply.
>
> > Some results are not adequately discussed, e.g. Theorem 5 and last paragraph on Pg 9.
>
> We will add a discussion to Theorem 5. Basically this is an instance where we show that abstention learning can specifically help overcome the difficulties of robustness even if there is no prior knowledge on the perturbation sets. This is for particular classes for which abstention learning is easy, i.e. disagreement coefficient is finite. We will also make sure to clean up the last paragraph and add a proper conclusion section to the whole paper instead.
>
>
> Regarding the realizability assumption:
>
> We agree that assuming that $\exists u^*\in\mathcal{U}$ and $h^*\in\mathcal{H}$ such that $\mathcal{L}^{u^*}_P(h^*) = 0$ is a strong assumption. However, as we show in Observation 1, not making any assumptions at all on $u^*$ makes the problem unsolvable (unless we have access to a perfect attack oracle). We further show in Theorem 2 that given access to a PAO, learning is possible in the agnostic case if we have a total order in $\mathcal{U}$. We also have a discussion in the supplementary material on agnostic learning with access to a PAO when $\mathcal{U}$ is a union of a finite number of total orders.
>
> In light of Observation 1, it is an interesting question to define a reasonable model of agnostic learning (agnostic also with respect to the perturbation class U) when we are neither given access to a PAO, nor allowed to abstain. We potentially have two ways to relax the realizability assumption. We can either relax the assumption that $u^*\in\mathcal{U}$ or that $h^*\in\mathcal{H}$. We study the latter in our present work, but it would be interesting to find an appropriate way of studying the former. We leave that for future work.

---

> > ### Comment · Reviewer_wkxJ · 2023-08-14
> >
> > Thanks for the detailed response. Investigation related to the MRERM oracle and potential for alternate ways of relaxing the realizability assumption sound like interesting further questions.

---

### Official Review · Reviewer_uGK4 · 2023-07-02

**Soundness:** 4 excellent
**Presentation:** 3 good
**Contribution:** 3 good
**Rating:** 7
**Confidence:** 3

**Summary:**

- This paper bridges the gap of PAC learning theory for adversarially robust learning between completely known and completely unknown perturbation across various settings.
- The authors introduce a notion of hypothesis class-induced partial ordering on the class of perturbation type which they use throughout the paper (setting 1).
- They show that in a realizable setting, hypothesis class with finite VC dimension and setting 1 are robustly learnable. They show that the same is not valid for agnostic setting without extra assumptions such as access to perfect attack oracle.
- They further show the existence of perturbation type and hypothesis class which cannot be robustly learned without abstention even with access to perfect attack oracle.
- They also investigate hypothesis classes with finite disagreement coefficients and present results for this setting.
- The define an $(\epsilon, \mathcal{H})$ cover for a class of perturbation types and use this definition to provide a generalization of their previous results.

**Strengths:**

The paper is well-written for the most part and provides a good explanation for the results. Furthermore, the claims seem mathematically sound and the authors have made good use of statistical learning theory to present their results. The results are interesting and close the gap between two known settings. They have also introduced some new notions such as the partial ordering of perturbation types with respect to hypothesis class and  $(\epsilon, \mathcal{H})$ cover for a class of perturbation types which are interesting concepts.

**Weaknesses:**

I liked the results and the general writing style of the paper. At times, the novelty seems to be a derivative of existing works with proofs being a clever extension of the proofs of existing settings. I believe this is bound to happen as the problem setting itself is an intermediary between two known settings.

The following is not a criticism but a remark - It would have been better to include some more mathematical proofs in a theoretical paper like this. However, I understand that space limitations sometimes prohibit this.

**Questions:**

Are there any perturbation models (other than norm balls) which satisfy the setting of this paper?

**Limitations:**

The authors have addressed the limitations adequately. In that, this paper doesn't propose any (efficient) optimization algorithms, it rather presents sample complexity guarantees with Oracle access to the solution of the optimization problems.

---

> ### Author Rebuttal · Authors · 2023-08-10
>
> Regarding other kinds of perturbation models:
>
> First, we should point out that “metric balls of given radius” already encodes many kinds of perturbation models we may want to use in practice. For example, consider perturbations only on a limited subset of features. Considering feature specific radii yield totally ordered classes, and considering differing collections of feature subsets yield finite unions of totally ordered perturbation type classes.
>
> Furthermore, note that our definition also allows one to define perturbation models where perturbations of different sizes are allowed at different data points. All we need is that if $u_i < u_j$, then $u_i(x)$ has a smaller radius than $u_j(x)$ for all $x$. This is more general than most models considered in the literature.
>
> If we want to go beyond metric balls, that can also be done, for example, by starting with any arbitrary shape and “expanding” it gradually.

---

> > ### Comment · Area_Chair_wmfg · 2023-08-18
> >
> > Dear authors,
> >
> > Your respondse has been noted.
> > The decision on your paper will be based on my discussion with the reviewers.
> > We will reach out to your should we require further clarifications.
> >
> > Regards,
> > AC

---

### Official Review · Reviewer_2e2f · 2023-07-06

**Soundness:** 4 excellent
**Presentation:** 4 excellent
**Contribution:** 3 good
**Rating:** 7
**Confidence:** 3

**Summary:**

This paper studies robust classification when the perturbation region an adversary can access is unknown to the learner, but where it is guaranteed it comes from a certain class of perturbations, which is known to the learner. The authors show various results when the learner has access to a perfect attack oracle (PAO) (which either returns a robust loss of 0, or an admissible perturbation on which the hypothesis is not robust).

For perturbation classes with total order, without access to the PAO, finite VC dimension is sufficient in the realizable setting, but not in the agnostic one. With access to the PAO, this becomes possible, by a reduction from agnostic to realizable robust learning.

When the perturbation class is a finite union of totally ordered perturbations, robust learning is also possible with the PAO (but in general, not without it). Removing structure on the perturbation class renders robust learning impossible (even with PAO).

The authors then consider robust learning with abstentions, where the learner cannot abstain on an unperturbed point, but can on a perturbed instance. They show that, in this set up,  robust learning is possible. They then relate robust learnability with the disagreement coefficient of a hypothesis class.

**Strengths:**

- Important question, nice set up that lies between unknown perturbation sets and models where the perturbation function is known a priori
- The results are interesting and draw on multiples different concepts (learning with abstentions, disagreement coefficients, previous robust learning results, etc.)
- The paper is clear and well-written
- Meets the technical standards of NeurIPS

**Weaknesses:**

- No conclusion section - please add one for the final version!
- MRERM: it seems to be quite a strong assumption to have access to this oracle? Especially since it returns the perturbation $u^*$! The existence of $u^*$ is fine, but that an algorithm can find it for any/most $\mathcal{H}$ seems quite strong. Could you expand on this, and give examples?

**Questions:**

- Proof of theorem 2: the quantity $n$ is not defined. Do you mean $n=2^m$, the set of all potential subsamples of $S$ ?

Comment:
- It would be worth adding on lines 124-125 and in the statement of Theorem 4 that the result holds for learning with abstentions.

Typos/etc:
- l.239 "implicite"
- "can not" vs "cannot" (multiple times throughout the text): e.g., "it can not be evaluated" means "it is possible that it is not evaluated", while "it cannot be evaluated" means "it is impossible to evaluate it" - quite different semantics!

**Limitations:**

yes

---

> ### Author Rebuttal · Authors · 2023-08-10
>
> Thank you for the suggestions! We will add a conclusion section and discuss current limitations, open questions and future directions in the final version. We will also fix the typos and other edits suggested (the comment about “can not” vs “cannot” was particularly interesting, and we had not thought about it before!). Thank you!
>
> Regarding MRERM:
> See the main comment addressed to all reviewers.
>
> Regarding the proof of Theorem 2: $n$ is simply the number of subsamples that are realizable with respect to some $u\in\mathcal{U}$. Thus $n\leq 2^m$.

---

> > ### Comment · Reviewer_2e2f · 2023-08-10
> > **Response**
> >
> > Thank you for the response and for the proposed changes!
> >
> > Regarding the MRERM clarification, I think a discussion in line with the one provided in the general rebuttal would be worth integrating in the final version of the paper (in case it wasn't in your plans already).

---

### Official Review · Reviewer_hPab · 2023-07-07

**Soundness:** 4 excellent
**Presentation:** 4 excellent
**Contribution:** 3 good
**Rating:** 6
**Confidence:** 3

**Summary:**

The authors present theoretical results on classification in a version of the PAC model where an adversary can perturb input instances.  Previous work assumed either a fixed perturbation type, known to the learner, or an unknown perturbation type.  This paper explores a middle ground where the perturbation type is a member of a fixed class of perturbation types.  The learner knows the class, but does not know which perturbation type within that class is being used.  For example, the perturbation class might correspond to balls with different radii.

The paper presents results on learning for a class U of perturbation types and a hypothesis class H.  It defines a partial order on perturbation types in U, defined with respect to H.  It considers variants of the learning model where the learner can abstain from predicting the label of a given example (which is counted as a misclassification error unless the example has been perturbed).  It also considers a variant of the learning model where the learner can has query access to a "perfect attack oracle," a type of oracle studied in previous work on learning with unknown perturbation sets.

The results in the paper are all with regard to sample size.  (Computational complexity is not considered.). The first two main results are as follows.  (1) When the class U is totally ordered with respect to H, and H has finite VC dimension, then learning in the realizable case is possible with a polynomial-sized sample (treating VC-dimension as a constant). (2) For the unrealizable case, when U is totally ordered wrt H, a polynomial-sized sample also suffices if the learner can also make polynomially many calls to a perfect attack oracle.

The next result considers a similar situation to that in (1) above, except in a more general setting where where U is a union of totally ordered subclasses.  In this setting, even with finite VC dimension and realizability, learning from (finitely many) examples is not possible, but can be achieved with a combination of a polynomial-sized sample and the ability to make polynomially many calls to a perfect attack oracle.

The remaining results are concerned with learning with abstentions.  The main results are (1) Learnability in the realizable case, with abstentions, when U is totally ordered and H has finite VC dimension (2) Learnability in the realizable case, with abstentions, when U is the class of all perturbation, when the class has finite VC dimension and finite disagreement coefficient.  A result is also given on learning with abstentions in the realizable case if there is a successful finite-disagreement-cover learner for (H,U).

The paper also contains simple negative results demonstrating the need for abstentions, total order, etc.

**Strengths:**

A comprehensive exploration of uncertain perturbation sets, bridging work on unknown perturbation sets on the one hand, and a single known perturbation set on the other hand.  The paper is well-written.

Discussion phase:  I raised my score from 5 to 6.

**Weaknesses:**

From the presentation in the main paper, it does not seem that the paper introduces new techniques or that the presented results are surprising.  It seems as if the techniques have already been used in previous work on perturbations and that the results are similar.  Overall, the work seems like a solid contribution, but lacking in real novelty.

**Questions:**

No specific questions.  It would be helpful to the reader to justify Definition 1, relating it to the possible behavior of an adversary.

**Limitations:**

I have no concerns in this area.

---

> ### Author Rebuttal · Authors · 2023-08-10
>
> > It would be helpful to the reader to justify Definition 1, relating it to the possible behavior of an adversary.
>
> Thanks for the feedback. We will add a justification in the final version. The intuition is that one way to handle adversarial attacks is to detect when an attack has happened and then abstain from making a prediction. The loss in Definition 1 allows the model to abstain when an attack has happened without penalty, but penalizes it if it makes a wrong prediction, thus incentivizing exactly the behavior described above.

---

### Official Review · Reviewer_o4ch · 2023-07-09

**Soundness:** 4 excellent
**Presentation:** 3 good
**Contribution:** 3 good
**Rating:** 6
**Confidence:** 5

**Summary:**

In adversarially robust PAC learning to test time attacks, usually we assume that the learner knows the perturbation function.
Montasser et al. ['21] studied a setting were the learner doesn't know the perturbation function, but can interact with it through some oracles. In this paper, the setting is in between, assuming that there is a class of possible perturbations $U$ known to the learner, and the performance of the learner is measured on the worst-case perturbation function in this set. For example, the perturbation can lie in a ball centered in the original input, but the norm isn't known.

The main contributions are as follows.

- Considering an order on $U$ w.r.t. $H$, finite VC is sufficient for learning in the realizable case, but not in the agnostic case.

- When the learner can interact with $U$ through a perfect attack oracle, and assuming some structure on $U$, finite VC is sufficient for learning. This is an improved result compared to the setting of an unknown perturbation function, where finite Littlestone dimension is sufficient, and $VC(H)\leq Lit(H)$ and the gap can be arbitrarily large.

- Introducing a setting where the learner can abstain, and showing that finite VC is sufficient for learning (assuming some structure on $U$).

- Assuming that $H$  has a finite disagreement coefficient (a parameter that is related to active learning), then $H$ can be learned with respect to every class of perturbations.

**Strengths:**

The in-between setting of robust learning suggested in this paper is natural and complements the picture of the other two settings (known $U$ or completely unknown $U$).
This paper studies thoroughly various scenarios, showing some limitations of this model, and when we can guarantee robust learning.
The results look correct to me.
I think that this paper could be of interest to the community of theoretical robust learning.

**Weaknesses:**

If I'm not missing anything, the technical contribution of the paper is moderate and mostly relies on standard ideas from PAC learning.

Many references on theoretical robust learning are missing. For example, H-consistency bounds for surrogate loss minimizers (ICML 2022), Multi-class H-consistency bounds (NeurIPS 2022), Theoretically grounded loss functions and algorithms for adversarial robustness (AISTATS 2023), Cross-Entropy Loss Functions: Theoretical Analysis and Applications (ICML 2023), A Characterization of Semi-Supervised Adversarially Robust PAC Learnability (NeurIPS 2022), Adversarially Robust PAC Learnability of Real-Valued Functions (ICML 2023), Sample complexity of robust linear classification separated data (ICML 2021)...and many more!

**Questions:**

-

**Limitations:**

I don't see any.

---

> ### Author Rebuttal · Authors · 2023-08-10
>
> Thank you for the additional references! We will make sure to include them in the final version!

---

### Author Rebuttal · Authors · 2023-08-10

We thank all reviewers for their positive reviews and appreciation of our work!

Two common themes that appeared in the reviews are addressed here, while more specific points have been addressed as replies to individual reviews.

Regarding the MRERM oracle:

Investigating when an MRERM oracle actually exists in practice is indeed interesting and it is on our todo list for future work. For some simple scenarios, such as, when $\mathcal{H}$ is halfspaces, and $\mathcal{U}$ is L2-balls, it is a simple convex optimization problem. In general, we could consider encoding the radius of the true $u\in\mathcal{U}$ using k bits. In this case, if we have access to RERM, then we can solve MRERM using a binary search over the radii, thus requiring $O(k)$ queries to the RERM oracle. In addition to exact MRERM oracles, it will also be interesting to investigate to what degree our learning techniques are robust to approximate versions of MRERM.

Regarding technical contributions:

The issue of not knowing the true perturbation radius has been a concern in the adversarially robust learning literature. Picking the wrong radius for adversarial training can lead to reduced accuracy, and thus it is important that we use the correct radius. Our work presents methods where there is no need to guess the radius. We show that it is possible to automatically optimize for the unknown, correct radius in much more general settings that such results had been known for before.
We agree that our proof techniques build up on previously known techniques. However, we find the fact that an algorithm like the above exists to be quite interesting. Since in many practical settings, the uncertainty is over a set of perturbation types that’s totally ordered, our algorithm covers many scenarios of interest. We are also first to prove that abstentions can be used to address the challenge of unknown perturbation sets.

---

### Decision · Program_Chairs · 2023-09-21

**Decision:**

Accept (poster)

**Comment:**

The reviewers agreed that this work is a solid contribution to robust learning theory.